# Facile Synthesis of Two Dimensional (2D) V_2_O_5_ Nanosheets Film towards Photodetectors

**DOI:** 10.3390/ma15238313

**Published:** 2022-11-23

**Authors:** Shaotian Wang, Liangfei Wu, Hui Zhang, Zihan Wang, Qinggang Qin, Xi Wang, Yuan Lu, Liang Li, Ming Li

**Affiliations:** 1Institutes of Physical Science and Information Technology, Anhui University, Hefei 230601, China; 2Key Laboratory of Materials Physics, Anhui Key Laboratory of Nanomaterials and Nanotechnology, Institute of Solid State Physics, Hefei Institutes of Physical Science, Chinese Academy of Sciences, Hefei 230031, China; 3State Key Laboratory of Pulsed Power Laser Technology, Anhui Laboratory of Advanced Laser Technology, Infrared and Low Temperature Plasma Key Laboratory of Anhui Province, National University of Defense Technology, Hefei 230037, China

**Keywords:** V_2_O_5_, two dimensional (2D) nanosheets, hydrothermal reaction, film, photodetectors

## Abstract

Most of the studies focused on V_2_O_5_ have been devoted to obtaining specific morphology and microstructure for its intended applications. Two dimensional (2D) V_2_O_5_ has the most valuable structure because of its unique planar configuration that can offer more active sites. In this study, a bottom-up and low-cost method that is hydrothermal combined with spin-coating and subsequent annealing was developed to prepare 2D V_2_O_5_ nanosheets film on quartz substrate. First, VOOH nanosheets were prepared by the hydrothermal method using V_2_O_5_ powders and EG as raw materials. Further, V_2_O_5_ nanosheets with an average lateral size over 500 nm and thickness less than 10 nm can be prepared from the parent VOOH nanosheets by annealing at 350 °C for 15 min in air. The prepared V_2_O_5_ nanosheets film was assembled of multiple nanosheets. The structural, morphological, microstructural and optical properties of the films were respective investigated by XRD, SEM, TEM and UV-Vis. The photodetector based on V_2_O_5_ nanosheets film shows good photoresponse with a response time of 2.4 s and a recovery time of 4.7 s.

## 1. Introduction

Vanadium oxides have attracted extensive attention for decades because of their unique physical and chemical properties [1]. Among them, layered vanadium pentoxide (V_2_O_5_) is thermodynamically the most stable oxide, which possesses the highest oxidation valence state among all vanadium oxides [2]. The rich crystallinity and morphology have made V_2_O_5_ a very interesting material for a broad range of applications, which include field emission [3,4], supercapacitor [5,6], lithium battery [7], transistor [8], and photodetector [9,10,11,12,13,14,15]. It is worth mentioning that V_2_O_5_ is a promising n-type semiconductor for photodetection due to its direct band gap of 2.2 to 2.7 eV [16,17]. Various V_2_O_5_ nanostructures, such as nanowire [4,18,19], nanosheet [20,21], nanorods [22,23,24], nanobelt [25], and nanoparticles [26] were studied as a photodetector. In particular, 2D nanostructures are ideal frameworks that provide a higher surface-to-volume ratio and a larger number of active sites, ultimately enhancing photoconductivity, showing promise for applications in optoelectronics [27,28,29,30,31], but, until very recently, there have been few reports on 2D nanostructured V_2_O_5_ [32].

Till now, some approaches have been explored for the synthesis of 2D V_2_O_5_ nanosheets, such as the hydrothermal method [33,34,35], additive-free ultrasonic method [36], exfoliation method [37,38], and pyrolysis [39]. Typically, Li et al. [40] prepared a 2D leaf-like V_2_O_5_ nanosheet with a thickness of 60–80 nm by a facile sol-gel method combined with freeze-drying technique followed with annealing in air. The morphology and size of the V_2_O_5_ nanosheets were controlled by adjusting the type of the solvent during a hydrothermal reaction, and the average thickness of the V_2_O_5_ nanosheet is about 74 nm [41]. Oxygen-deficient V_2_O_5_ nanosheets with a thickness of 30~40 nm were prepared by the drop-casting method using dimethyl sulfoxide solution containing vanadyl acetylacetonate, then thermally annealed at a certain temperature [42]. Song et al. prepared mesoporous V_2_O_5_ nanosheets with a thickness ranging from 20 nm to 50–70 nm, and the mesoporous structure produces a high specific surface area and possesses an abundance of oxygen vacancies [33]. However, the number of reports on preparing 2D V_2_O_5_, especially those with a thickness below 10 nm that has exotic electronic properties, is still very small. Even though a few reports claimed to have obtained 2D V_2_O_5_ with a thickness less than 10 nm [43,44,45,46,47], the mechanical exfoliation methods and the addition of various organic solvents are inefficient for massive production. Consequently, to prepare 2D V_2_O_5_ with the desired property, there is still a space to develop a facile and scalable synthetic process that provides an excellent control of the morphology, defects, and thickness of V_2_O_5_.

In this study, we report a general and facile hydrothermal combined with rapid annealing method to successfully prepare 2D V_2_O_5_ nanosheets films. Firstly, 2D VOOH nanosheets that have a lateral size exceeding 500 nm and a thickness less than 10 nm can be simply prepared by hydrothermal treatment of V_2_O_5_ and ethylene glycol (EG). Secondly, stable and homogeneous VOOH dispersion was made by ultrasonic processing in ethanol without using any surfactant or modifiers, and then, the dispersion was spin-coated onto quartz glass substrates to form a VOOH film. Finally, the 2D V_2_O_5_ nanosheets films were obtained by rapid annealing of VOOH at 350 °C in air. By contrast, our preparation technique is low cost and easy to operate as compared to the conventional techniques and can be scaled up to produce large amounts of V_2_O_5_ samples. Meanwhile, metal semiconductor–metal (MSM)-based photodetectors are widely used in emerging communication systems due to their unique structure [48,49,50]. Therefore, we fabricated the MSM photodetector based on 2D V_2_O_5_ nanosheets film and studied its optoelectronic properties systematically. The response time and a recovery time of the synthesized 2D V_2_O_5_ nanosheets photodetector are 2.4 s and 4.7 s, respectively, which are faster than other reports of ~60–70 s and ~10 s (Appendix A). The particular photoelectronic properties of V_2_O_5_ nanosheets make them candidate materials for photodetectors, which have bright application prospects.

## 2. Materials and Methods

### 2.1. Preparation of 2D V_2_O_5_ Film

V_2_O_5_ powders (Tianjin Zhiyuan Chemical Reagent Co., Ltd., Tianjin, China), deionized water (18.2 MΩ.cm, MERCK MILLIPORE pure water system), anhydrous ethanol (AR, Sinopagic reagent), ethylene glycol (EG, AR, ≥99.0%, national reagent) as raw materials, all starting materials were of analytical pure grade and can be used directly without any purification. First, 0.5 g V_2_O_5_ powders were added into fully mixed and stirred with 15 mL glycol and 20 mL deionized water, and then added to the 45 mL Teflon-lined autoclave. The temperature and time of hydrothermal reaction were 200 °C and 24 h, respectively. After cooling in the reactor, the reacted powder was repeatedly centrifuged with ethanol to obtain VOOH nanosheets powders. Finally, the VOOH nanosheets powders were directly added into ethanol with an ultrasonic treatment for 30 min to obtain good dispersion, even though there was no surfactant added.

The stable and homogeneous VOOH dispersion was then applied onto a 2 × 2 cm quartz substrate using a spin-coating method at a rotational acceleration of 800 rpm/s^2^ and a rotational speed of 1500 rpm/s. The thickness of the VOOH film can be controlled by changing the spin-coating times, which can be transformed to 2D V_2_O_5_ nanosheets films by a rapid annealing at 350 °C for 15 min.

### 2.2. Device Fabrication and Characterization

The V_2_O_5_ nanosheets film was fixed on the homogenizer, and AZ5214 photoresist was applied to the sample film uniformly with a disposable pipette, with homogenization performed at a low speed of 600 rpm for 10 s and at a high speed of 7000 rpm for 60 s. The obtained photoresist thickness was about 1 μm. Then, the sample was baked on the heating table at 110 °C for 4 min. The electrodes were fabricated by standard photolithography techniques via UV lithography (UV exposure for 10 s, optical power of 15 mW), followed by Ti/Au thermal evaporation (15/50 nm) and a later lift-off process.

Morphology of the product was characterized by scanning electron microscope (Helios nanolab 600i SEM with NPGS system). The phase and composition were identified using an Empyrean S3 X-ray diffractometer (XRD, PANalytical X′Pert). The microstructure was characterized by transmission electron microscope (TEM, JEOL JEM-2010), and the selected area electron diffraction (SAED) was obtained by JEMARM 200F. X-ray photoelectron spectroscopy (XPS) was used to analyze the composition and oxidation states of vanadium oxide (Thermo ESCALAB 250Xi). A Raman system (FST2-Ahdx-DZ) equipped with a 532 nm laser source was used to record the Raman spectra. A UV–Vis spectrometer (UV-3600 spectrophotometer Shimadzu ISR-260) was used to obtain the absorption spectra. The V_2_O_5_ nanosheets film photodetector was observed with an optical microscope (OM) (WMJ-9688). The photoelectric performances of the fabricated device were investigated with a Keithley 4200A-SCS.

## 3. Results and Discussion

The schematic diagram in Figure 1a displays the preparation process of 2D V_2_O_5_ nanosheets films. Briefly, the VOOH nanosheet was first prepared by hydrothermal, followed by ultrasonic processing in ethanol to convert it into VOOH dispersion, which was then spin-coated on a quartz substrate to prepare VOOH nanosheets film, and finally, the V_2_O_5_ nanosheets film was obtained by subsequent rapid annealing at 350 °C.

Both the VOOH and the V_2_O_5_ crystalline phase were confirmed by XRD characterization, as seen in Figure 1b, in which the diffraction peak of the hydrothermal product can be indexed as VOOH (JCPDS card no. 74-1877), where the (020) crystalline plane has a clear preferred orientation [51,52,53]. Simultaneously, a series of XRD peaks of the post-annealed sample allowed us to identify the orthorhombic phase of V_2_O_5_ (JCPDS card no. 41-1426). Three distinct diffraction peaks of (001), (101) and (400) clearly demonstrate the polycrystalline and high purity features of 2D V_2_O_5_ film [33,54]. Figure 1c shows the SEM image of the VOOH film, from which one can see that the discrete nanosheets with lateral size of over 500 nm assembled together to form a nanosheets film. After annealing, the obtained V_2_O_5_ film preserved the nanosheet morphology and film configuration as well as that of VOOH (Figure 1d). The magnified SEM image (Appendix A) gives further insight that the thickness of the stacked nanosheets is less than 10 nm. In addition, the surface of the post-annealed nanosheets showed slight cracks due to volume changes during the oxidation process. Ultimately, the VOOH nanosheets are oxidized to V_2_O_5_ nanosheets after annealing corresponds to the following equation:(1)$$2\mathrm{VOOH}+O_{2}\overset{350\;{}^{\circ}C}{\to }V_{2}O_{5}+H_{2}O$$

From a crystallographic perspective, as shown in Figure 2a–c, the orthorhombic crystalline V_2_O_5_ has a laminar framework structure (space group: Pmmn, a = 1.1516 nm, b = 0.3566 nm, c = 0.4372 nm), with each V_2_O_5_ layer consisting of a VO_5_ square pyramid that shares sides and corners, and the different layers of the VO_5_ pyramids are connected by weak van der Waals forces. In the same layer of the VO_5_ pyramid, oxygen and vanadium atoms are covalently bonded to form a complete crystal structure. To further reveal the microstructure of as-prepared 2D V_2_O_5_ nanosheets, they were characterized using high-resolution transmission electron microscopy (HRTEM). Prior to HR-TEM analysis, V_2_O_5_ film was exfoliated from the substrate as a stand-alone nanosheet and subsequently dispersed in ethanol, followed by ultrasonic treatment of the solution and dropping it on a TEM grid. After performing the drying process, the grid was then taken for TEM analysis. As the HRTEM shows in Figure 2d, the lattice striations can be clearly observed, which suggests that the V_2_O_5_ nanosheets have good crystallinity. In addition, the lattice spacings of 0.289 nm and 0.349 nm are consistent with the interplanar distances of (301) and (201) planes that belong to the orthorhombic V_2_O_5_, respectively. At the same time, according to the SAED pattern shown in Figure 2e, the V_2_O_5_ polycrystalline phase was confirmed. Figure 2f exhibited the energy dispersive X-ray (EDX) spectrum, which suggested that the as-prepared V_2_O_5_ nanosheets included 37.27% and 62.73% of V and O, respectively. Figure 2g–i show the TEM image and EDX elemental mapping of the selected V_2_O_5_ nanosheet, and the results show that the prepared samples have uniform spatial distribution of V and O.

The elemental composition and chemical states of the prepared samples were identified using X-ray photoelectron spectroscopy (XPS). Figure 3a presents the survey scan XPS spectrum of the as-prepared sample, which clearly shows the introduction of O and V elements (charge corrected to C 1s = 284.8 eV). Asymmetric peaks were seen in the low-binding energy portions of the V2p_3/2_ peak and V2p_1/2_ peak, indicating the existence of multiple oxidation states of V_2_O_5_. As shown in Figure 3b, the spectra of the V2p_3/2_ oxidation state can be separated to V^4+^ (516.06 eV) and V^5+^ (517.45 eV) through the Lorentzian–Gaussian fitting. Meanwhile, the spectra of the V 2p_1/2_ oxidation state were separated to V^4+^ (523.69 eV) and V^5+^ (525.07 eV) [55]. The V^5+^ state at the high-binding energy corresponds to V_2_O_5_, while the V^4+^ state responds to VO_2_, which indicates the gap state [56]. Figure 3c presents the O_1s_ core level XPS spectrum, and the peaks at 532.17 eV and 530.28 eV belong to adsorbed oxygen and lattice oxygen, respectively [57]. Our XPS spectrum matches well with those of the V_2_O_5_ nanosheets reported in the literature [33,56,57]. Meanwhile, the Raman technique is a powerful method to identify the chemical composition of the material. Figure 3d presents the Raman spectrum of V_2_O_5_ nanosheets in the wave number range of 100–1100 cm^−1^, which displays the eight positional signals corresponding to the phonon patterns reported previously: 145 cm^−1^ (B_1g_/B_3g_), 197 cm^−1^ (A_g_/B_2g_), 284 cm^−1^ (B_1g_/B_3g_), 304 cm^−1^ (A_g_), 405 cm^−1^ (A_g_), 481 cm^−1^ (A_g_), 700 cm^−1^ (B_1g_/B_3g_), and 993 cm^−1^ (A_g_) [17,18,58]. The Raman spectrum matches well with the V_2_O_5_ reported in the literature [18]. The orthorhombic crystalline structure of V_2_O_5_ nanosheets is thus further confirmed by the Raman spectrum. It is worth mentioning that two main Raman peaks at low-frequency locations of 145 and 197 cm^−1^ correspond to [VO_5_]–[VO_5_] vibrations, which originated from the lattice-bending vibration, indicating the long-range order layered structure of V_2_O_5_ nanosheets. Furthermore, the peaks at 405 and 284 cm^−1^ are bending vibration of the V=O (terminal oxygen) bonds, and the peaks at 304 and 481 cm^−1^ are bending vibrations of V_3_-O (triply coordinated oxygen) bonds and V-O-V (bridging doubly coordinated oxygen), respectively. Moreover, the peak in 700 cm^−1^ belongs to the asymmetric stretching of V_2_-O (doubly coordinated oxygen) bonds, and the intensity feature at 993 cm^−1^ belongs to the stretching vibrations of V=O (terminal oxygen) vanadyl bonds in V_2_O_5_, which is distinguished from other vanadium oxide in the Raman spectrum [32,59,60].

For further research on the optical features of the V_2_O_5_ nanosheets film as photodetector, the absorbance spectrum is recorded as presented in Figure 4a. The spectra show the major absorption bands of V_2_O_5_ nanosheets film at around ~350–450 nm, which indicates that the synthesized V_2_O_5_ nanosheets film can be utilized as a photodetector covering the range of visible light. As presented in Figure 4b, the optical band gap of the synthesized V_2_O_5_ was calculated to reach 2.73 eV by using the Tauc plot. The absorbance spectrum of V_2_O_5_ nanosheets in Figure 4 closely matches with that reported in the literature [20,61]. At the same time, our optical band gap of the synthesized V_2_O_5_ is close to that of V_2_O_5_ nanosheets reported in the literature, which is 2.4 eV [21].

Figure 5a illustrates the device fabrication workflow. The electrode patterns were formed on V_2_O_5_ nanosheets film by ultraviolet lithography (UV exposure for 10 s, optical power of 15 mW) firstly, and then, Ti/Au (15/50 nm) were deposited on the film by thermal evaporation and later lift-off procedures. The distance between the Ti/Au electrodes is about 15 μm. The OM images before and after deposition of electrodes are shown in the insets of Figure 5a.

The photodetecting performances of the V_2_O_5_ nanosheets film were investigated. Figure 5b illustrates the schematic diagram of a V_2_O_5_ nanosheets film photoelectric device. We investigated the photodetecting performances under the visible light laser irradiation at a wavelength of λ = 405 nm. Figure 5c presents the comparisons of the I-V relationship under dark conditions and in laser irradiation of a varying power intensities from 31.8 to 503.2 mW cm^−2^. The results show that the obtained I-V curves observed show nonlinearity, which thus indicates the generation of a Schottky barrier between the V_2_O_5_ nanosheets film and the Ti/Au electrode, while the photocurrent increases sharply under illumination, especially under high voltage bias. The phenomenon that the slope for the I-V curve increases with rising power density demonstrates that the V_2_O_5_ nanosheets film photodetector is photosensitive. Then, the I-T curves were measured under different power density laser irradiation at a bias voltage of 1V. As illustrated in Figure 5d, with the rise of the power density (31.8 to 503.2 mW cm^−2^), the photocurrent increases gradually. Figure 5e shows the logarithmic relationship between light intensity and photocurrent at a wavelength of 405 nm, wherein the experimental data of the photocurrent (black discrete solid square dots) and fitted curves (red solid line) were taken as a function of light density (*P*). The curve can be fitted by a formula of *I*_ph_~*P*^*θ*^, where *I*_ph_, *P* and *θ* represent the photocurrent (*I*_ph_ = *I*_light_ − *I*_dark_), the incident light power density, and the index of the power law, respectively. The quantitative analysis of the experimental data clearly shows that the equation was fitted to be *I*_ph_~*P*^0.955^. As shown in the fitted curve, the photocurrent gradually increases sub-linearly with increasing light density at a voltage of 5 V. In accordance with the results and reports of previous studies, the deviation of the slope from the ideal value of *θ* = 1 is caused by complex factors and processes, such as defects in spin-coated prepared nanosheets film and loss of photoexcitation carriers [11,62,63]. Therefore, we believe that *θ* = 0.955 is mainly caused by defects in V_2_O_5_ nanosheets film.

The time-dependence photoresponse behavior is examined by turning on/off the incident light at 20 s intervals at an optical power density of 503.2 mW cm^−2^ (V_bias_ = 5 V). It was performed for several cycles, and the result is presented in Figure 5f; the current in the device exhibits fine repeatability and steady response in the case of periodic on and off laser irradiation, corresponding to the periodic variation of laser light. Meanwhile, it can be observed that the dark current increases slightly with the increase of the illumination time at room temperature. The primary factor can be ascribed to the local thermal effect on the device caused by laser illumination, resulting in an increasing dark current with time [11]. Figure 5g depicts a zoomed-in view of a circle in Figure 5f; the rise time and decay time are 2.4 and 4.7 s, respectively.

For further evaluation of the performance of V_2_O_5_ nanosheets film applied as photodetector, as the function of optical power density, the *R* (responsivity), *D** (detection rate) and *EQE* are calculated. The *R*, *D**, and *EQE* can be denoted as *R_λ_* = *I_ph_*/*PS*, *D** = *R_λ_S^1/2^*/*(2eI_dark_)^1/2^*, and *EQE* = *R_λ_hc*/*eλ*, respectively, where *S* and *P* denote the photodetectors’ effective area and optical power density, *h*, *c*, and *e* denote Planck’s constant, velocity of light, and electron charge, respectively [62,64]. At a power density of 503.2 mW cm^−2^ at a bias voltage of 5 V, the calculated responsivity and detectivity of the V_2_O_5_ nanosheets film photodetector were 2.86 mA W^−1^ and 2.058 × 10^6^ Jones, severally. Additionally, the *EQE* of the V_2_O_5_ nanosheets film photodetector is calculated to be 0.8773%. Appendix A has summarized the key parameters of nanostructured V_2_O_5_ photodetectors. The V_2_O_5_ nanosheets photodetector in this work has short response time that was found to outperform the same type of V_2_O_5_ nanosheets photodetector [10,18,20,21,24,65].

## 4. Conclusions

In conclusion, we report a facile and economical strategy to successfully prepare 2D V_2_O_5_ nanosheets film by using VOOH nanosheets produced by hydrothermal combined with spin-coating and subsequent annealing treatment. The prepared V_2_O_5_ nanosheets film was assembled from a plurality of nanosheets with lateral dimensions greater than 500 nm and thickness of about 10 nm. The prepared V_2_O_5_ nanosheets were characterized using SEM, XRD, TEM, XPS, Raman, and UV-Vis spectroscopy. The photodetector based on 2D V_2_O_5_ nanosheets film shows good optoelectronic performance with a response time of 2.4 s and a recovery time of 4.7 s. Our results show that V_2_O_5_ nanosheets film is simple to prepare, low cost, and can be prepared in large areas at scale, making it a highly suitable candidate material for developing photodetectors.

## Figures and Tables

**Figure 1 materials-15-08313-f001:**
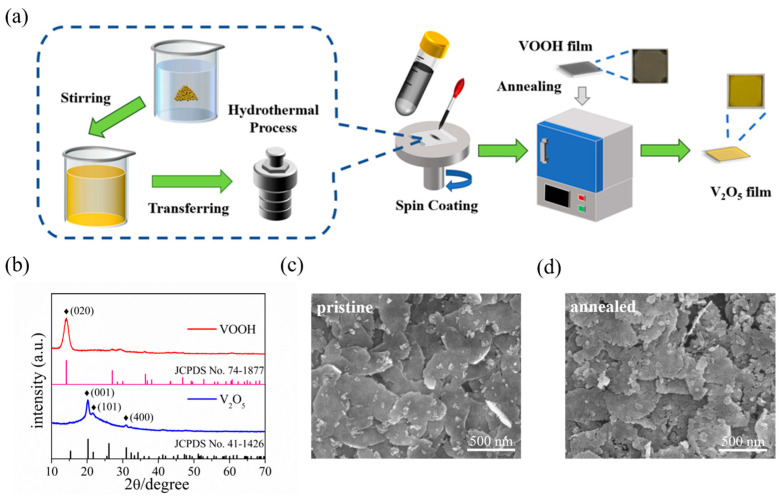
(**a**) Schematic illustration of the synthesis route of V_2_O_5_ nanosheets film in this work. (**b**) XRD patterns of VOOH and V_2_O_5_. (**c**) SEM image of the VOOH nanosheets film. (**d**) SEM image of the V_2_O_5_ nanosheets film.

**Figure 2 materials-15-08313-f002:**
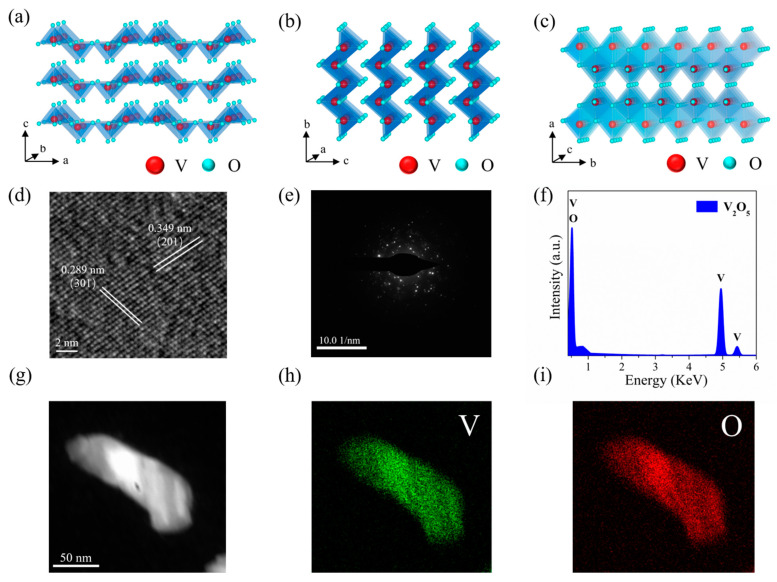
(**a**–**c**) Schematic diagram illustrating the crystal structure of V_2_O_5_. (**d**) HRTEM image of the V_2_O_5_ nanosheet. (**e**) SAED pattern of the V_2_O_5_ nanosheet. (**f**) EDX spectrum of V_2_O_5_ nanosheet. (**g**) TEM image of the V_2_O_5_ nanosheet. (**h**,**i**) Elemental mapping images of the V_2_O_5_ nanosheet.

**Figure 3 materials-15-08313-f003:**
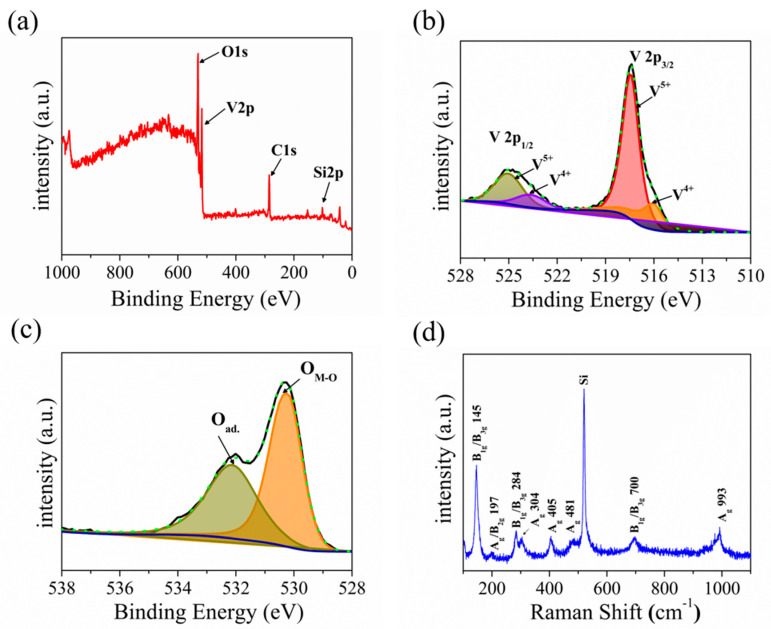
(**a**) XPS survey spectrum of the V_2_O_5_. (**b**,**c**) High−resolution V_2p_, O_1s_ XPS peak and corresponding fitting spectrum. (**d**) Raman spectrum of the of V_2_O_5_.

**Figure 4 materials-15-08313-f004:**
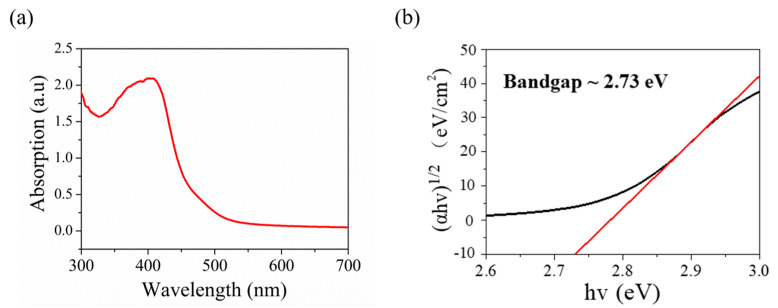
(**a**) UV−vis spectroscopy of synthesized V_2_O_5_ nanosheets. (**b**) Tauc plot showing the optical band gap of 2.73 eV.

**Figure 5 materials-15-08313-f005:**
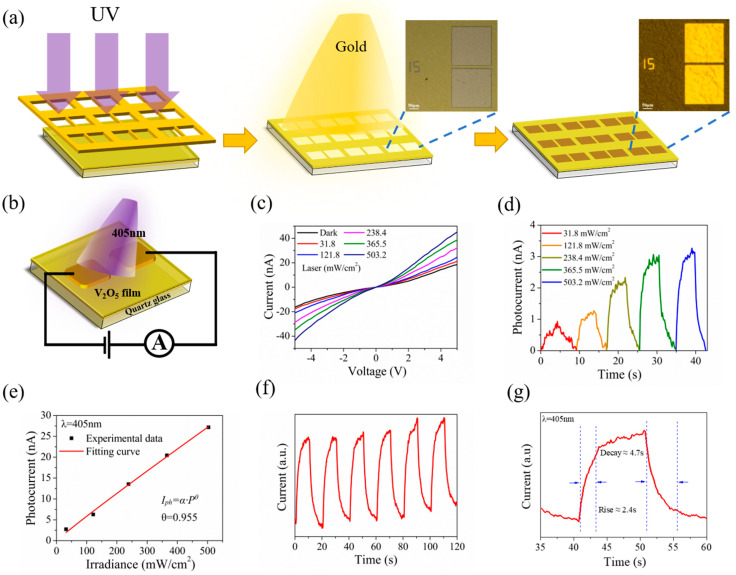
(**a**) Schematic diagram of electrode deposition on V_2_O_5_ film. Inset: The optical image. (**b**) Schematic of the V_2_O_5_ nanosheets film-based photodetector. (**c**) I−V curves of the device in the dark and under illumination with different intensities. (**d**) Light intensity−dependent photoresponse at V_bias_ = 1 V. (**e**) Photocurrent as a function of laser power intensity. (**f**) Time−resolved photoresponse of the device recorded for a power density of 503.2 mW cm^−2^. (**g**) Response rate of photodetector acquired from one magnified circle of response with rising time 2.4 s and decay time 4.7 s.

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
