# Peer review of "Facile Synthesis of Two Dimensional (2D) V2O5 Nanosheets Film towards Photodetectors"

_materials, 2022, doi:10.3390/ma15238313_

Round 1

Reviewer 1 Report

Reviewer #1: In this manuscript, the authors reported ‘Facile Synthesis of Two dimensional (2D) V2O5 Nanosheets Film Towards Photodetectors’. The structural, morphological, micro- structural and optical properties of the films were respectively investigated by XRD, SEM, TEM and UV-Vis. From this side, this manuscript is interesting to the readers. The topic is important in this field. However, there are several issues needed to be improved and revised before the possible publication in this high-quality journal.  I suggest some corrections, which should be taken into consideration before the publication.
Further comments:

1.      A careful English language revision is strongly recommended in order to remove some mistakes and misspellings appearing through the entire text of manuscript. For the topic of 2D materials, some recent reports, such as   https://doi.org/10.1016/j.mssp.2019.03.017, https://doi.org/10.1016/j.optmat.2022.112097, https://doi.org/10.1088/1361-6528/aba4cd, etc are valuable for being referred to. It is suggested the authors to check their manuscript carefully and thoroughly to avoid some typical mistakes and mistypes.

2.       The referee suggests enriching the discussion based on the experimental results, which will be very important for the readers in the relative field. More recent literature is suggested to be included.

3.      The photo detector performance based on the V2O5 should also compared with 2D synthesized by other methods.

4.      In the experiment, the protocol on the fabrication of photo detector should be added in this manuscript and photodetector parameters likes responsivity, detectivity and EQE   are compare with other photodetector in tabulation.

Reviewer 2 Report

  1. What is freedring technique at line 49
  2. In figure 1B, XRD peaks do not match the JCPDS card. Extra peak is there in VOOH. What is that? 
  3. Figure 2 d, the lattice spacing is presented in 4 decimal places. What is the accuracy of the measurement?  
  4. Compare the Raman Shift peaks with the literature 
  5. Please explain the behavior of figure 5f, Time-resolved photodetection of the device at 20 and  90 seconds.  Extra going down and going up of the curve, what does it infer 
  6. How do authors say that film thickness is 10 nm? Cross-sectional TEM/SEM is required for this
  7. The results have not been compared with the results of the literature
  8. Do authors believe that V2O5 nanosheets PD film PD can be commercialized? What needs to be improved?
  9. The literature says various kinds of PD (Enhanced photoconductivity performance of microrod-based Sb2Se3 device, Broadband Photodetectors Based on Kesterite Thin Films, Low bias operated, fast response SnSe thin-film Vis-NIR photodetector on glass substrate using one-step thermal evaporation technique) exists. Why use V2O5 as a photodetector.
  10. It says in figure S2a, i.e., contacts are ohmic due to the sub-linear range of I-V. However, after that same contacts are said to be Schottky by this in figure 5 (b). Please confirm figures Sa and b.

  11. Y axis scale (numbering) with figure 4 (b) unit must be added.

  12. What is achieved here, at least, results compared with V2O5 literature?

    Whys is it better than others?   

Reviewer 3 Report

This work “Facile synthesis of two dimensional (2D) V2O5 nanosheets film towards photodetectors” demonstrates a photodetector based on V2O5 nanosheet using hydrothermal synthesis process. The results are well organized and great interest to the community. However, there are several points need to be clarified before publishing this manuscript.

1.     The device fabrication process as shown in Figure 5a should be revised. There is a mistake regarding the optical images before and after metal deposition. After lift-off step, why did metal show a black color?

2.     The electrical characteristics of V2O5 devices in Figure S2a and Figure 5c are not consistent. The author mentioned about the ohmic contact (Figure S2a), while it shown Schottky contact (Figure 5c). I suggest calculating the Schottky barrier height (SBH) from the temperature-dependent measurement. Are they the sample device or any comment for this part?

3.     From the literature, the CB of V2O5 is estimated around 5 eV, what is main metal contact type to form with V2O5 in this case? Ti or Au? Author should provide the band diagram as well with the estimation of SBH.

4.     The response rate of photodetector in Figure 5g seems not to be as from 5f. Please have look through it again in the time range between 45-70s.

Reviewer 4 Report

The comments and suggestions are given below for the paper titled: Facile Synthesis of Two dimensional (2D) V2O5Nanosheets Film Towards Photodetectors

In the presented paper, a bottom-up  and low-cost method that is hydrothermal combined with spin-coating and subsequent annealing was developed to prepare 2D V2O5nanosheets film on quartz substrate. The obtained results seem interesting in the field of sensors and optoelectronics devices. Moreover, the paper is well written and free from scientific mistakes. However, there are some issues that should be addressed in order to improve the manuscript. The main comments are listed below:

1- Some minor writing typos and sentences should be corrected and improved.

2- The benefits of MSM-based photosensors as building blocks for the emerging optical communication systems have not been discussed suitably in the manuscript. In this regards, please review the following papers and provide some comments about this aspect in the Introduction Section: DOI: 10.1109/JSEN.2019.2920815; https://doi.org/10.1016/j.jallcom.2020.158242; https://doi.org/10.1016/j.mssp.2020.105331.

3- According to Fig.4, I think that the proposed sensor can be used as broadband photodetector (UV-Vis) device, which is widely used for imaging applications. This point should be discussed in the paper. In this context, I suggest referring to: https://doi.org/10.1016/j.optmat.2022.112578 ; https://doi.org/10.1016/j.jallcom.2020.158242  

4. I suggest adding a comparison table with other published results, including some Figures of Merit such as: Response and recovery time; responsivity and Ion/I dark ratio...

In summary, the achieved work showcases significant contributions. Thus, I would recommend with Minor revision this manuscript for the possible publication in the Journal.

Round 2

Reviewer 2 Report

Some language correction is still necessary 

Reviewer 3 Report

The revised version is significantly improving the quality of manuscript, all questions are clearly clarified. Therefore, I suggest accepting this study to publish in the Materials.

Author Response

Thank you for your recognition!
